# An Approach to Integrating Sentiment Analysis into Recommender Systems

**DOI:** 10.3390/s21165666

**Published:** 2021-08-23

**Authors:** Cach N. Dang, María N. Moreno-García, Fernando De la Prieta

**Affiliations:** 1Department of Information Technology, HoChiMinh City University of Transport (UT-HCMC), Ho Chi Minh 70000, Vietnam; 2Data Mining (MIDA) Research Group, University of Salamanca, 37007 Salamanca, Spain; mmg@usal.es; 3Biotechnology, Intelligent Systems and Educational Technology (BISITE) Research Group, University of Salamanca, 37007 Salamanca, Spain; fer@usal.es

**Keywords:** sentiment analysis, deep learning, recommender system, natural language processing

## Abstract

Recommender systems have been applied in a wide range of domains such as e-commerce, media, banking, and utilities. This kind of system provides personalized suggestions based on large amounts of data to increase user satisfaction. These suggestions help client select products, while organizations can increase the consumption of a product. In the case of social data, sentiment analysis can help gain better understanding of a user’s attitudes, opinions and emotions, which is beneficial to integrate in recommender systems for achieving higher recommendation reliability. On the one hand, this information can be used to complement explicit ratings given to products by users. On the other hand, sentiment analysis of items that can be derived from online news services, blogs, social media or even from the recommender systems themselves is seen as capable of providing better recommendations to users. In this study, we present and evaluate a recommendation approach that integrates sentiment analysis into collaborative filtering methods. The recommender system proposal is based on an adaptive architecture, which includes improved techniques for feature extraction and deep learning models based on sentiment analysis. The results of the empirical study performed with two popular datasets show that sentiment–based deep learning models and collaborative filtering methods can significantly improve the recommender system’s performance.

## 1. Introduction

With the explosion of blogs, forums, and online social networks, differing opinions about a particular topic can be easily found from millions of users. For example, users can discuss their current experiences, share their points of view on a specific fact, or praise or complain about a product that they have just bought. With a vast amount of available online data, sentiment analysis—a method to categorize text-based opinions to determine a user’s attitude—can help gain better understanding of the attitudes, opinions and emotions of the public in several domains such as business, government, and biomedicine. Several studies are summarized and discussed in [1] regarding the benefits of sentiment analysis in obtaining feedbacks and determining the interests and opinions of customers.

Recommender systems, first developed in the mid-1990s and based on users’ ratings and preferences, have expanded widely in recent decades. They are now especially important in the realms of e-commerce, media, banking, and utilities. This type of system is used by Amazon to suggest preferred products for customers, by YouTube to suggest related videos on the auto-play function, and by Facebook to recommend people and webpages to connect and follow.

Sentiment analysis can be beneficial to recommender systems. A sample of this can be found in the work of Preethi et al. [2], in which a cloud-based recommender system uses recursive neural networks to analyze sentiments of reviews in order to improve and validate restaurant and movie recommendations. Along with behavioral analysis, sentiment analysis is also an efficient tool for commodity markets [3].

Social media data has been exploited in different ways to address some problems, especially associated with collaborative filtering approaches [4]. Methods in recommender systems are based on information filtering, and they can be classified into three categories: content-based; collaborative filtering (CF); and hybrid. Sparsity and gray-sheep problems are two of the main reasons CF methods do not provide the reliability required in some recommender systems [5]. In particular, when only sparse ratings data is available, sentiment analysis can play a key role in improving recommendation quality. This is because recommendation algorithms mostly rely on users’ ratings to select the items to recommend. Such ratings are usually insufficient and very limited. On the other hand, sentiment-based ratings of items that can be derived from reviews or opinions given through online news services, blogs, social media or even the recommender systems themselves, are seen as capable of providing better recommendations to users. Sentiment-based models have been exploited in recommender systems to overcome the data-sparsity problem that exists in conventional recommender systems. Hence, integrating sentiment in recommender systems may significantly enhance the recommendation quality.

In this study, we propose a recommendation method that combines sentiment analysis and collaborative filtering. The method is implemented in an adaptive recommender system architecture in which techniques for feature extraction and deep learning-based sentiment analysis is included. The results of the empirical study performed with two popular datasets show that combining deep learning-based sentiment analysis and collaborative filtering methods significantly improve the recommender system’s performance.

The rest of this paper is organized as follows. Section 2 presents background information and provides a literature review in this research area. Section 3 describes the methodology for recommender systems. Section 4 outlines the results and discussion, and Section 5 offers the main conclusion.

## 2. Background and Related work

Sentiment analysis is very useful in a wide range of application domains, including business, government, and education. Application of sentiment analysis in recommender system has also been the focus of extensive research. In this section, we start by presenting background information and reviewing the literature to offer an up-to-date overview of how sentiment analysis has been applied in recommender systems.

### 2.1. Sentiment Analysis

Sentiment analysis can be performed on three levels of extraction: the sentence level; the document level; and the aspect or feature level. It is a process of extracting information about an entity and automatically identifying any of the subjectivities of that entity. The aim is to determine whether text generated by users conveys their positive, negative, or neutral opinions. Three approaches currently exist to address the problem of sentiment analysis [6]: lexicon-based techniques; machine-learning-based techniques; and hybrid approaches. Lexicon-based techniques are divided into two approaches: dictionary-based and corpus-based [7]. They were the first to be used for sentiment classification. Machine learning-based techniques [8] that have been proposed for sentiment analysis include traditional techniques and deep learning techniques. The hybrid approaches is the combination of machine learning and lexicon-based approaches [9]. Sentiment lexicons regularly play a key role in most of these strategies. Figure 1 illustrates a taxonomy of deep learning-based methods for sentiment analysis.

Deep learning techniques can provide better results than traditional techniques. Different kinds of deep learning models can be used for sentiment classification, including CNN, DNN, and RNN. These models address classification problems at the document level, sentence level, or aspect level. In addition, some approaches that combine two models are introduced [10,11,12,13,14,15,16]. The CNN enhanced by SVM [10,11,12], CNN with RNN [13,14,15,16] showed enhanced results.

The hybrid models can increase the accuracy for sentiment analysis in comparison to a single model performance. In this study, we combine deep learning techniques for sentiment analysis. The resulting hybrid deep-learning models for sentiment analysis, which combine LSTM networks [17] and CNN [18], are built and tested on two datasets containing reviews.

### 2.2. Recommender Systems

A recommender system intends to provide personalized recommendations about products or services to support decision making in the continuous increase of online information. Several systems have been developed and applied in three main domains: business, government, and education, across eight categories: e-government, e-business, e-commerce/e-shopping, e-library, e-learning, e-tourism, e-resource services and e-group activities [20]. E-commerce has widely applied recommender systems to suggest additional products for customers to choose from among the multiple products available. A filtering technique has improved systems for presenting personalized choices [21].

The most common methods used for recommender systems may be grouped into three categories: content-based; collaborative filtering (CF) and hybrid recommender systems [22]. These techniques vary depending on the types of social media data that are used. Lu et al. [20] analyzed typical recommender systems and effectively identifies the specific requirements for recommendation techniques in the domain. This work also directly motivates and supports researchers and practitioners to promote the popularization and application of recommender systems in different domains.

Content-based recommender systems: Content-based methods make use of characteristics of items and users’ profiles. User profiles are created by mining content information about items accessed over the web by users, such as product attributes. Content-based recommender systems filter items based on the content-based similarity measures between items in the catalog and items that users have previously consumed, accessed, or rating positively. Therefore, a user receives recommendations of items like those that previously have been of interest. The utility of an item for a user can be a derivative done after a quantitative analysis of the metadata of the item.

Collaborative filtering-based recommender systems: Collaborative filtering is a technique that can filter out items that a user might like based on reactions by similar users. It works by searching in a large group of people and finding a smaller set of users with tastes like those of a particular user. It looks at the items they like and combines them to create a ranked list of suggestions. We need data that contains a set of items and a set of users to perform with recommender algorithms. While working with such data, the matrix consists of the reactions given by a set of users to certain items from within a set of items. Each row would contain the ratings given by a user, and each column would contain the ratings received by an item.

Hybrid recommender systems: Hybrid approaches take advantage of any kind of item and user information that can be extracted or inferred from web systems, social media, or other sources. Hybrid approaches are implemented by deployment individually as well as by accumulating rankings and predictions and then building a general consolidative model that resolves the common problems in recommender systems.

Each recommendation approach has advantages and limitations; for example, Collaborative Filtering has sparseness, scalability and cold-start problems [5,23,24]. A sparseness problem occurs when we have a vast amount of data. A scalability problem occurs when the rating data is missing. When a user or an item is added to the system the cold-start problem appears. Combining sentiment analysis with recommendation methods can help solve these problems. Figure 2 shows the categories of deep learning applied to information retrieval and recommender system research.

### 2.3. Related Work

Recommender systems can be improved in a variety of ways. In [4], social tag embedding is used in a collaborative filtering approach in which user similarities based on both tag embedding and ratings are combined to generate the recommendations. Recommender systems have also benefited from sentiment analysis. An example of this can be found in the work of Preethi et al. [2], where recursive neural networks were applied to analyze sentiments in reviews. The output was used to improve and validate restaurant and movie recommendations of a cloud-based recommender system. Along with behavioral analysis, sentiment analysis is also an efficient tool for commodity markets [3]. Wang et al. [26] combined a hybrid recommender system and sentiment analysis to optimize the preliminary list and obtain the final recommendation list. Kumar et al. [27] proposed a hybrid recommender system by combining collaborative filtering and content-based filtering with the use of sentiment analysis of movie tweets to boost up the recommender system.

Rao et al. [28] designed a recommender system that contains the user list and item list with user reviews. Using the sentiment dictionaries, the researchers divided the items into three categories: brand, quality, and price. They leveraged sentiment dictionaries to calculate sentiment of a particular user on item/product. Gurini et al. [29] adopted a different approach to describe a user recommender system for Twitter. Their work emphasized the use of implicit sentiment analysis in order to improve the performance of the recommendation process. They defined a novel weighting function that considers sentiment, volume, and objectivity related to the users’ interests.

In yet another approach, Osman et al. [30] presented an electronic product recommender system based on contextual information from sentiment analysis. Because ratings are usually insufficient and very limited, they constructed a contextual information sentiment model for a recommender system by making use of user comments and preferences. In a similar way, Contratre et al. [31] also proposed a recommender process that includes sentiment analysis of textual data extracted from Facebook and Twitter in order to increase conversion by matching product offers and consumer preferences. We can find similar combinations in other studies [32,33,34].

In addition, Rosa et al. [35] used a sentiment intensity metric to build a music recommender system. Users’ sentiments are extracted from sentences posted on social networks and the recommendations are made using a framework of low complexity that suggests songs based on the current user’s sentiment intensity. The research by Osman, Nurul Aida, and Shahrul [36] addressed the data-sparsity problem of recommender systems by integrating a sentiment-based analysis. Their work was applied to the Internet Movie Dataset (IMDb) and Movie Lens datasets, but improvements in sentiment analysis have been made since the paper was published. Rayan et al. [37] also tried to improve recommendations by addressing the data-sparsity problem. They proposed a smart recommender system based on methods of hybrid learning that integrate the most effective and efficient learning algorithms. These methods switch among content-based and collaborative filtering, identify the user context with the integration of dynamic filtering, and finally learn the profiles.

Several research teams [26,27,33,38,39,40] introduced the techniques for applying sentiment analysis in recommender systems. The techniques that are applicable for performing the analysis of sentiments include support vector machines (SVM), Convolutional Neural Networks (CNN), Recurrent Neural Networks (RNN), and deep neural networks (DNN).

Recommender systems rely on explicit user ratings, but this is not feasible in an increasing number of domains. Moreover, when explicit ratings are available, the trust and reliability of the ratings may limit the recommender system. When we have a large number of reviews and comments on these items, analyzing sentiments in that text to obtain implicit feedback in addition to traditional ratings for items, is useful and helps to improve the recommendations to users. The above studies use sentiment analysis in recommendation methods, but most studies have used traditional sentiment techniques or a sole deep learning model.

In this study, we will apply new feature extraction techniques and hybrid deep-learning methods for sentiment analysis exploiting the advantages of BERT, in order to incorporate sentiments into recommendation methods as additional feedback and thus improve the performance and the reliability of recommender systems.

## 3. Methodology

In this section, the proposed recommender system is presented. It is based on a recommendation method that combines collaborative filtering and sentiment analysis. The aim is to improve reliability of the recommendations to the user by combining sentiment analysis of reviews or comments of users with traditional recommendation methods. The architecture of this system is illustrated in Figure 3. The architecture makes it easy to configure the modules and their interactions, allowing the application to be composed by choosing from supported techniques and methods.

The architecture has two separate parts, one part in charge of generating the sentiment models and the other part to provide recommendations to a given user making use of the models previously generated. The reviews’ data were preprocessed and used to conduct and train sentiment-based hybrid deep-learning model. Then, a user-based (user-user) collaborative filtering method is combined with sentiment-based models for rating prediction.

### 3.1. Input Data and Preprocessing

Sentiment analysis requires that the text training data are cleaned before being used to induce the classification model. Text cleaning is a preprocessing step that removes words or other components that lack relevant information, and thus may reduce the effectiveness of sentiment analysis. After cleaning, the text data can be split into individual words, which are transformed into their base form by lemmatization, and then converted into numerical vectors by using methods such as word embedding or TF-IDF. Both word embedding and TF-IDF are used as input features of deep learning algorithms in nature language processing [41].

For the deep learning approaches, word embedding representations have performed significantly better than the TF-IDF representation of all features and feature selection algorithms [1,42]. In this research, we used BERT to transform text data to word embedding. Word embedding [43] is a type of word representation that maps each word into a vector of real values in such a way that words with similar meanings have a similar representation. Value learning can be done using neural networks. BERT is a language model for nature language processing, and it was published by researchers at Google AI Language in 2018 [44]. BERT was developed after Word2vec, and includes some advances over Word2vec, such as support for out-of-vocabulary (OOV) words.

### 3.2. Conduct and Train Sentiment-Based Hybrid Deep-Learning Models

We used the combination of several successful approaches. We start by using a pre-trained BERT model to create the feature vectors. We then vary the order of the CNN and LSTM models used in the next stages: BERT → CNN → LSTM or BERT → LSTM → CNN. The final stage of the model uses a ReLu activation function. We labeled the reviews with one value of an ordinal scale of five classes (very negative; negative; neutral; positive; and very positive), analogous to the explicit ratings, to train and validate the result of sentiment analysis.

Figure 4 visualizes the process of the hybrid methodology for sentiment analysis. A pre-trained BERT model was used in our experiments as a feature extractor to generate input data for the proposal of hybrid models. The reviews data were fed into the BERT model to generate the feature vectors, which are then input to the hybrid models that perform the classification. The next step combines CNN and LSTM deep learning models, which are used because of their good performance on sentiment analysis [1], as well as to take advantage of the two network architectures when performing sentiment analysis on data in different domains. The final stage is classification. We use the activate function of Relu instead of Sigmoid because of the high convergence.

### 3.3. Proposed Recommendation Method

The proposed recommendation method is a user-based collaborative filtering approach that considers explicit ratings and sentiment analysis extracted from users’ reviews. We tested Singular Value Decomposition (SVD), Non-Negative Matrix Factorization (NMF), and SVD++ (a derivative of SVD) as collaborative filtering methods. The objective is to achieve better predictive accuracy because of the addition of implicit feedback information provided by the sentiment.

Results from the CF recommendation method and sentiment analysis were combined to generate a rating and used to create a list of recommendations.

Given a rating matrix Rm × n (ℕ) for training, where m is the number of users and n is the number of items, rij∈Rm × n denotes the rating of user ui  on item ij.

The rating of user $u_{a}$ on item $i_{j}$ in the test set is predicted as follows:(1)$$pr_{aj}=\;\beta *pr\_mf_{aj}+\left(1-\beta \right)*pr\_sent_{aj}$$
where:
*p*$r\_mf_{aj}$: Rating for user $u_{a}$ and item $i_{j}$ predicted by Matrix Factorization methods (SVD, SVD++, and NMF) without using sentiments.$pr\_sent_{aj}$: Rating for user $u_{a}$ and item $i_{j}$ predicted by using the sentiment model.β parameter used to adjust the importance of each term of the equation.

As shown in Algorithm 1, we used pseudocode to describe how to compute $pr\_sent_{aj}$. As mentioned above, hybrid sentiment models are used for classifying each “review” in one of five possible classes. These classes are converted into sentiment scores from 1 to 5 analogous to ratings. First, for each user $u_{a}$, we find all items that user $u_{a}$ already rated and the sentiment score of the corresponding review matches the explicit rating. And second, for each item $i_{j}$, we also find all users who already rated item $i_{j}$ and item $i_{k}$ (found in the first step) in the training set and their review scores also match the explicit ratings.
**Algorithm 1.** Rating prediction based on sentiment for user $u_{a}$ and item $i_{j}$.1.Function sentiment_ratingPred (user $u_{a}$, item $i_{j}$) {2.*//This function is used to obtain the*$pr\_sent_{aj}$*term of Equation (1)*3.* //Step 1:*4.  FOR each item $i_{k}$ in the training set:5.   IF user $u_{a}$ already rated item $i_{k}$ AND review score matches rating THEN6.    Add $i_{k}$ to list of items I;7. *//The result of this step is a set of m items* 
$I=\left\{i_{1},i_{2},\ldots i_{m}\right\}$
8.  *//Step 2:*9.  FOR each user $u_{b}$ in the training set:10.   FOR each item $i_{k}$ in the set of items I:11.    IF user $u_{b}$ already rated item $i_{j}$ AND user $u_{b}$ already rated item $i_{k}$
      AND their review scores match ratings 12.     Add user $u_{b}$ to list of users U;13. *//The results of this step is a set of n users* 
$U=\left\{u_{1},u_{2},\ldots u_{n}\right\}$
14.  *//Step 3:*15.  IF length(U)>0 THEN16.   FOR each user $u_{i}$ in the set of user U:17.    Compute $s_{a,i}$= sim (user $u_{a}$, user $u_{i}$) by applying cosine metric;18.    Add $s_{a,i}$ to S;19.  *//The result is a set of n similarity values* 
$S=\left\{s_{a,i}\right\}$
20.   Set the K value to select the K nearest neighbors using S;21.   Compute the predicted rating $pr_{aj}$ by applying the Equation (2);22.  
$pr\_sent_{aj}=pr_{aj}$
23.   Return $pr\_sent_{aj}$;24.  ELSE25.   Return 0;26.} 

Next, two lists of data, including items and users which are created from step 1 and step 2, are used for predicting user $u_{a}$ rating on each item $i_{j}$. To do that, we compute the similarity between users by applying the cosine metric. Then, we apply Equation (2) for rating prediction based on user similarity. The ratings of the k most similar users are used to estimate the preferences of the active user $u_{a}$ about the item $i_{j}$ that he/she has not rated.
(2)$$pr_{aj}=\bar{r_{a}}+\frac{\sum_{i=1}^{K}Sim\left(u_{a},u_{i}\right)\left(r_{ij}-\bar{r_{i}}\right)}{\sum_{i=1}^{K}|Sim\left(u_{a},u_{i}\right)|}$$
where $r_{ij}$ is the rating that user $u_{i}$ gives to item $i_{j}$ respectively; $\bar{r_{a}}$ and $\bar{r_{i}}$ are the average ratings of user $u_{a}$ and user $u_{i}$, respectively; and $Sim\left(u_{a},u_{i}\right)$ is the similarity between the active user $u_{a}$ and his neighbor user$\;u_{i}$, which would be obtained by using the cosine metric (Equation (3)). In our case, the neighbors of user $u_{a}$ are users who have rated the same items as user $u_{a}$ in a similar way or the score of their reviews on the same items are similar.
(3)$$Sim\left(u_{a},u_{i}\right)=\frac{\sum_{j=1}^{n}r_{aj}\;r_{ij}}{\sqrt{\sum_{j=1}^{n}r_{aj}^{2}\;}\sqrt{\sum_{j=1}^{n}r_{ij}^{2}\;}}$$

## 4. Experiments and Results

In this section, we present the experiments conducted to evaluate the performance of the proposed approach to recommender systems. In particular, we used two well-known datasets, Amazon Fine Food Reviews and Amazon Movie Reviews, in order to validate the proposal. The results are shown and discussed in Section 4.2. The metrics used to evaluate the reliability of rating predictions were Root-Mean-Square Error (RMSE), Mean Absolute Error (MAE) and Normal MAE (NMAE). In addition, Mean Reciprocal Rank (MRR), Mean Average Precision (MAP) and Normalized Discounted Cumulative Gain (NDCG) were used for evaluating top-N recommendations. Accuracy, Area Under Curve (AUC), and F-score were the metrics used to evaluate the performance of the two hybrid deep-learning models for sentiment analysis through all experiments. Because the F-score is the average of the F-score of each class, with weighting depending on the average parameter in the multi-class and multi-label case, we used the average parameter with ‘weighted’ value to calculate the metrics for each label and to find their average weighted by support.

The configuration of related parameters, hardware devices, and the necessary library facilities was carried out before performing the experiments, such as echo = 5, and k-fold = 5. In particular, we used Google Colab Pro with GPU Tesla P100-PCIE-16GB or GPU Tesla V100-SXM2-16GB [45], Keras [46] and Tensorflow [47] libraries. We also used the implementation of the SVD, NMF, and SVD++ algorithms provided by the Surprise library (http://surpriselib.com/, accessed on 10 December 2020).

### 4.1. Dataset

We chose the datasets based on availability and accessibility criteria. Moreover, we considered that they are widely accepted by the research community. These datasets are shown in Table 1 and described below:Amazon Fine Foods Reviews comprise reviews of fine foods from Amazon [48]. Each review includes product and user information, as well as the rating, and the plaintext review given by each user to each product he/she rated. The data span a period of more than 10 years, including 568,454 reviews with 256,059 users and 74,258 products up to October 2012.Amazon Movie Reviews consists of movie reviews from Amazon [48]. Each review also includes product and user information, ratings, and plaintext reviews. It covers a period of more than 10 years as well, including 7,911,684 reviews with 889,176 users and 253,059 products up to October 2012.

### 4.2. Results and Discussion

We performed experiments with two different settings without/with sentiment analysis. In the former, recommendations are based on recommender system methods without sentiment while in the second, the result of performing sentiment analysis on the reviews is incorporated to the recommendation process. We tested two hybrid deep-learning models for sentiment analysis: CNN and LSTM as well as LSTM and CNN, referred to as C-LSTM, L-CNN, respectively.

As presented in Figure 4, we adopt a pre-trained BERT model to vectorize each plaintext review. The obtained vector is then fed into C-LSTM or L-CNN followed by the fully connected layer. Finally, ReLU is stacked on the top of the classifier. The output of the sentiment classifier is exploited for recommendation. Table 2 and Figure 5 present the experimental results of sentiment classification. The results show that the performances of the hybrid models are encouraging, with accuracy and F-score over 80% and AUC over 84%, respectively. These models will be applied to predict sentiment rating before being combined with recommendation methods.

To validate our recommendation approach, we compared the performance of three widely used CF recommendation methods in their traditional form as baseline and the same methods improved with our proposal involving use of sentiment analysis of reviews. The comparative study was conducted for both rating prediction and item recommendation (recommendation of top-N lists).

Table 3, Table 4 and Table 5 show the results of MAE, RMSE and NMAE measures for rating prediction on both dataset food and movie reviews. They were calculated based on SVD, NMF and SVD++ algorithm with and without using sentiment analysis. Beta (β) parameter is used to adjust the importance of the recommendation result without and with sentiment in the Equation (1). Figure 6, Figure 7 and Figure 8 illustrate the comparative results obtained from the recommender with sentiment analysis on different values of the β parameter against those obtained from the recommender without sentiment analysis.

The results show that RSME, MAE, and NMAE yielded by the approach that combines CF with sentiment analysis are better than the error rates yielded by traditional CF methods without sentiment on all algorithm in all β values. We found that the best results of the proposal are obtained with β = 0.3.

Regarding the type of datasets, Amazon Movie Reviews provided better results than those of Amazon Fine Foods reviews. For example, MAE measured with SVD++ is 0.577 with β = 0.3; RMSE measured with SVD++ is 0.8577 with β = 0.3; and NMAE measured with SVD++ is 0.1443 with β = 0.3.

Figure 9 and Figure 10 illustrate the comparison of the sentiment-based methods with the L-CNN and the C-LCTM against non-sentiment-based methods on Amazon Fine Foods Reviews and Amazon Movie Reviews. The values with sentiment are obtained with β = 0.3. We found that C-LSTM and L-CNN provide similar results. In addition, the sentiment-based approach provides better results on Amazon Movie reviews.

For all algorithms applied to two datasets, the combined proposal provides lower error rates. For example, the sentiment-based approach on Amazon Movie Reviews with β = 0.3 when L-CNN is used for sentiment analysis provided the following percentage improvement with SVD: 12.29% in RMSE; 27.01% in MAE; and 6.75% in NMAE. With the above results, we see that the sentiment model helps to improve the predicted ratings. Instead of just using explicit rating, the predictive model now considers the aspect of analyzing reviews of related items and users. Because more information is available in the new recommendation method, we get better than usual results.

In addition to proving that the proposed method performs better in predicting ratings, we also checked the performance for top-N recommendations. MRR, MAP, and NDCG rank-based metrics have been computed. The results obtained for N = 5 are given in Table 6 and Table 7, and in Figure 11 and Figure 12. SVD, NMF, and SVD++ values, respectively, with L-CNN and C-LSTM sentiment models are obtained when applied on β = 0.7.

The values of MRR, MAP, and NDCG show that the proposed method also improve top-N recommendations. In the case of the Amazon Foods Review with β = 0.7 and C-LSTM sentiment model, the increase in MAP was 0.58% (SVD), 0.26% (NMF), and 0.36% (SVD++) over without sentiment. Regarding NDCG, the increase was 0.25% (SVD), 0.15% (NMF), and 0.06% (SVD++) over without sentiment. The value of MRR was increased on 0.24% (SVD), 0.25% (NMF), and 0.09% (SVD++).

Three algorithms (SVD, NMF, and SVD++) were tested in two ways, with explicit ratings only, and combining explicit ratings with sentiment extracted from reviews. In most cases, the combined approach with sentiments from two classification models (C-LSTM and L-CNN) on food and movie reviews datasets gave better results. However, the improvement for top-N recommendation is not as significant than the achieved for rating prediction.

In general, the sentiment-based methods proposed in this work provide better results than those based only on explicit ratings. These improvements have occurred in both data sets used in the study. General summaries of the results achieved in the experiments referenced earlier are discussed below:We presented and evaluated a recommendation approach that integrates sentiment analysis and collaborative filtering methods.Two datasets, Amazon Fine Foods Review and Amazon Movie Review, are used for evaluation. Each plaintext review is vectorized by using the pre-trained BERT model.Two hybrid sentiment classification models, CNN-LSTM and LSTM-CNN, are used for extracting sentiments from reviews, which are incorporated as implicit feedback into the recommender system models.We applied SVD, NMF, and SVD++ recommendation methods following the user-based CF approach.Accuracy, F-score, and AUC were computed for validating the sentiment classification models.The evaluation of the recommendation method was performed for rating prediction and top-N recommendation. RMSE, MAE, and NMAE were the metrics used in the first case, and MRR, MAP and NDCG were the metrics used in the second case.The sentiment-based proposal increased the recommendation reliability in comparison to traditional, rating-based recommendation methods on the two datasets.

## 5. Conclusions

In this paper, we have proposed an application of sentiment analysis in recommender systems that is based on hybrid deep-learning models and collaborative filtering on online social networks. The system architecture presented in this work, can integrate a variety of techniques that have been proposed to perform recommendations, including the preprocessing strategy, hybrid deep-learning models for sentiment analysis and methods for recommender systems. The architecture can be used to develop a recommender system in the context of social networks that take advantage of sentiment analysis performed on user opinions and reviews in the network. We conducted experiments with reviews of food and movies. Based on such experiments, we demonstrate the utility and applicability of our approaches in producing personalized recommendations on online social networks.

The results show that the joint use of deep learning-based sentiment analysis and collaborative filtering methods significantly improves the performance last ones. This is achieved through the exploitation of additional information from user reviews/comments data. Its integration into the traditional recommendation methods makes the recommender system more reliable and capable of providing better recommendations to users.

As a future work, we plan to explore other application domains to ensure that the proposed architecture can be generalized to efficiently solve similar problems. We will also consider researching new sentiment analysis techniques, such as graph convolutional networks, for a potential improvement of this aspect.

## Figures and Tables

**Figure 1 sensors-21-05666-f001:**
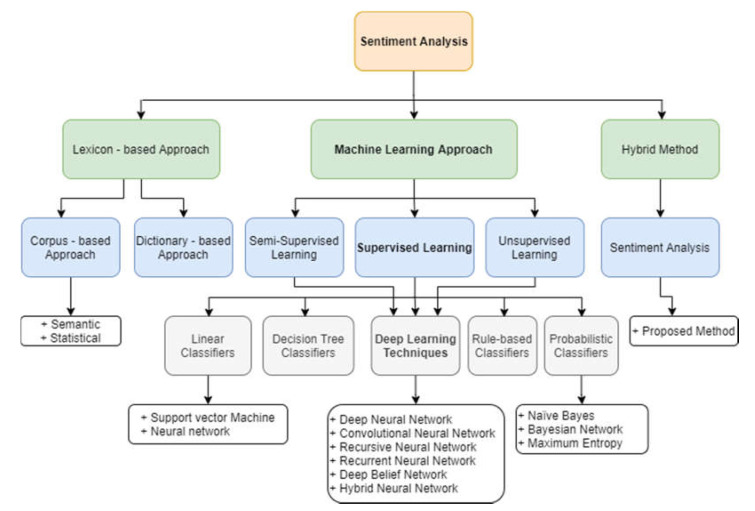
Taxonomy of sentiment analysis techniques. Source: [6,19].

**Figure 2 sensors-21-05666-f002:**
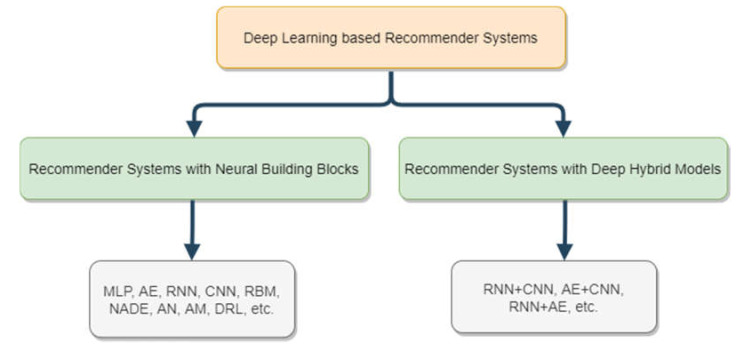
Categories of deep neural network-based recommendation models [25]. Multilayer Perceptron (MLP); Auto Encoder (AE); Convolutional Neural Network (CNN); Recurrent Neural Network (RNN); Restricted Boltzmann Machine (RBM); Neural Autoregressive Distribution Estimation (NADE); Adversarial Networks (AN); Attentional Models (AM); Deep Reinforcement Learning (DRL).

**Figure 3 sensors-21-05666-f003:**
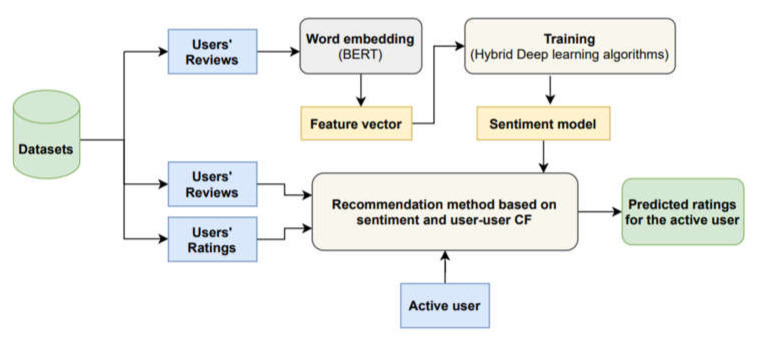
An architecture application in a recommender system.

**Figure 4 sensors-21-05666-f004:**
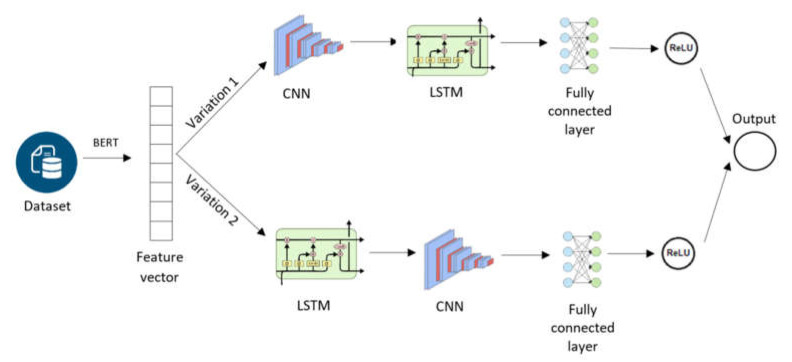
Process of hybrid methodology for sentiment analysis.

**Figure 5 sensors-21-05666-f005:**
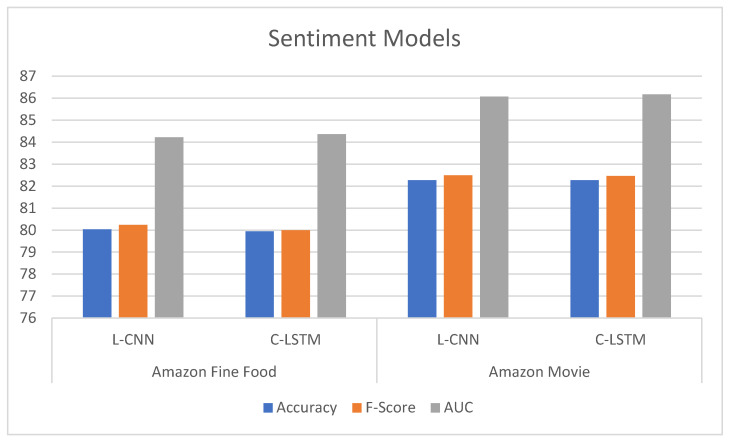
Sentiment performance of hybrid deep-learning models using BERT.

**Figure 6 sensors-21-05666-f006:**
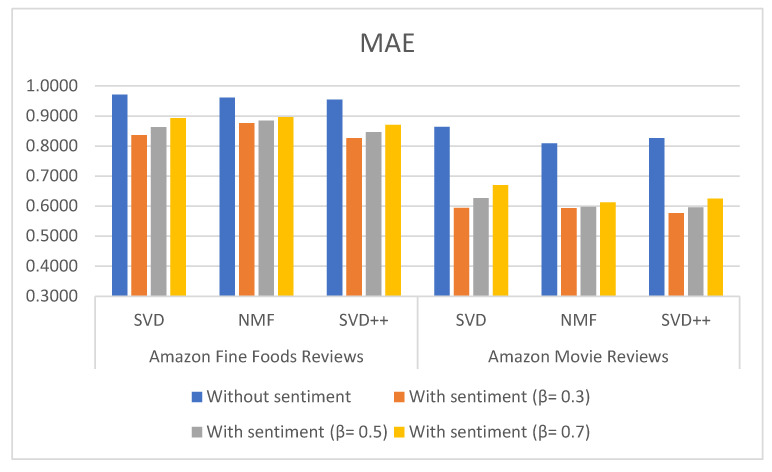
MAE measures comparison for different types of method and datasets using L-CNN sentiment model.

**Figure 7 sensors-21-05666-f007:**
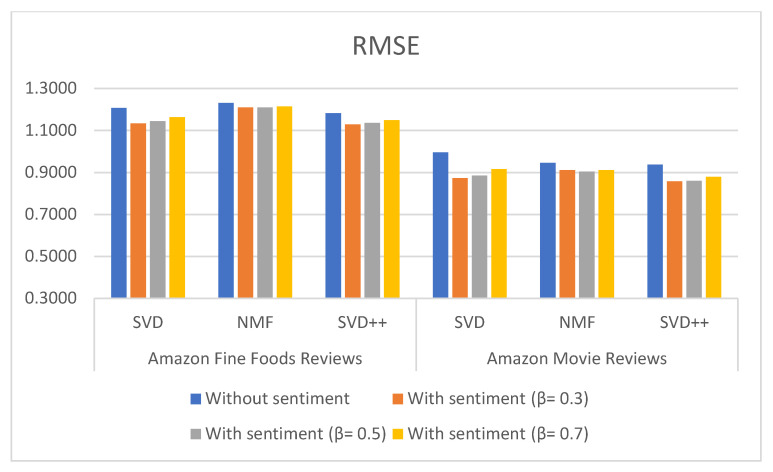
RMSE measures the comparison for different types of methods and datasets using L-CNN sentiment model.

**Figure 8 sensors-21-05666-f008:**
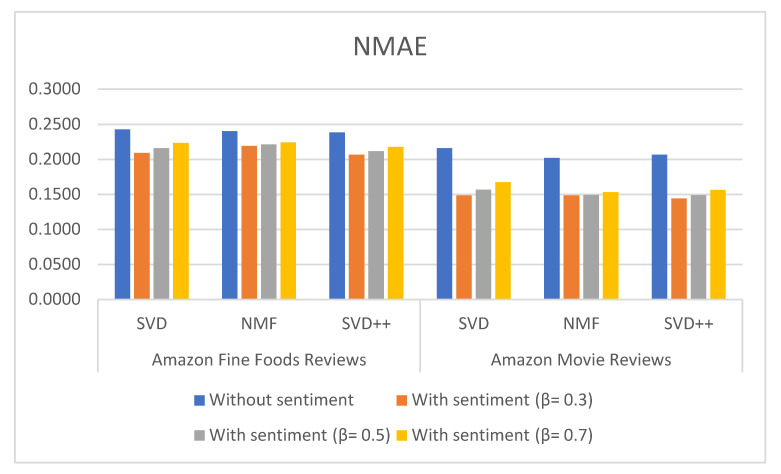
NMAE measures the comparison for different types of methods and datasets using L-CNN sentiment model.

**Figure 9 sensors-21-05666-f009:**
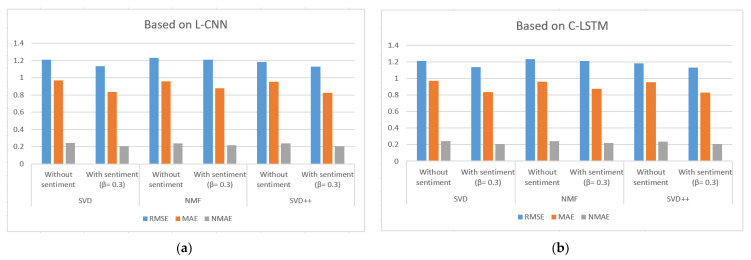
Comparison of the sentiment-based methods with the L-CNN model (**a**) and the C-LSTM (**b**) and β = 0.3 against non-sentiment-based methods on Amazon Fine Foods Reviews.

**Figure 10 sensors-21-05666-f010:**
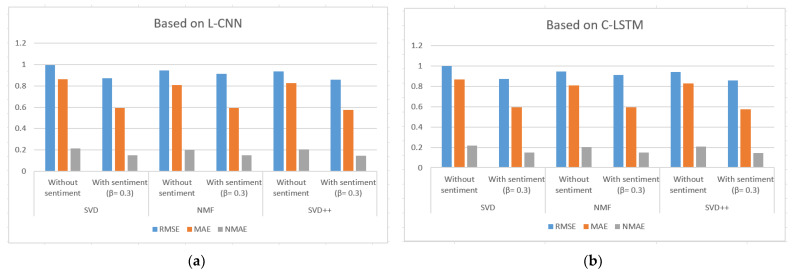
Comparison of the sentiment-based methods with the L-CNN model (**a**) and the C-LSTM (**b**) and β = 0.3 against non-sentiment-based methods on Amazon Movie Reviews.

**Figure 11 sensors-21-05666-f011:**
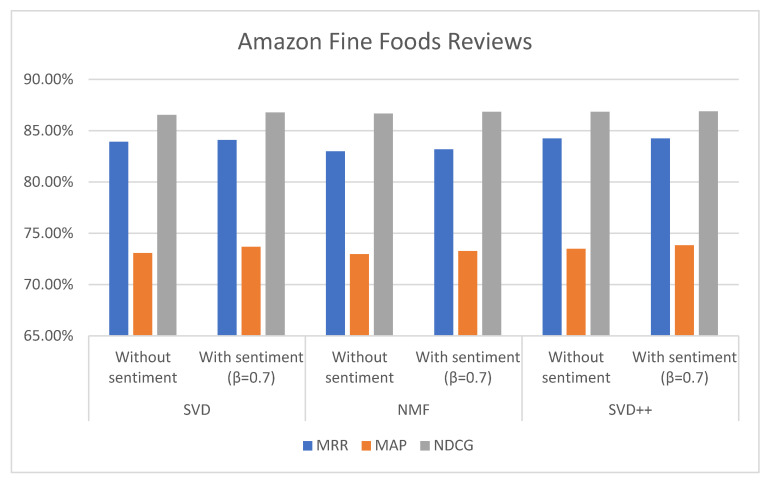
MRR, MAP, and NDCG values without and with L-CNN sentiment model on Amazon Fine Foods Reviews with β = 0.7.

**Figure 12 sensors-21-05666-f012:**
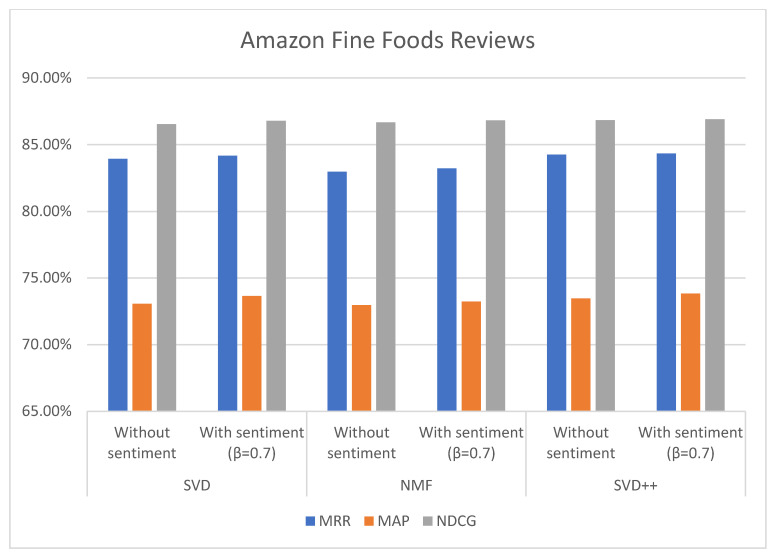
MRR, MAP, and NDCG values without and with C-LSTM sentiment model on Amazon Fine Foods Reviews with β = 0.7.

**Table 1 sensors-21-05666-t001:** Statistics of the datasets.

#	Amazon Fine Foods Reviews	Amazon Movie Reviews
Number of reviews	568,454	7,911,684
Number of users	256,059	889,176
Number of products	74,258	253,059
Users with > 50 reviews	260	16,341
Average no. of words per review	56	101
Timespan	October 1999–October 2012	August 1997–October 2012

**Table 2 sensors-21-05666-t002:** Sentiment performance of hybrid deep-learning models.

Measures	Amazon Fine Foods Reviews		Amazon Movie Reviews	
	L-CNN	C-LSTM	L-CNN	C-LSTM
Accuracy	80.04%	79.95%	82.27%	82.27%
F-Score	80.24%	80.00%	82.49%	82.46%
AUC	84.22%	84.36%	86.07%	86.17%

**Table 3 sensors-21-05666-t003:** MAE values without and with L-CNN sentiment analysis model.

#	Amazon Fine Foods Reviews			Amazon Movie Reviews		
	SVD	NMF	SVD++	SVD	NMF	SVD++
Without sentiment	0.9706	0.9608	0.9540	0.8644	0.8087	0.8266
With sentiment (β = 0.3)	0.8365	0.8762	0.8263	0.5943	0.5936	0.5770
With sentiment (β = 0.5)	0.8634	0.8846	0.8470	0.6268	0.5976	0.5959
With sentiment (β = 0.7)	0.8933	0.8964	0.8707	0.6701	0.6125	0.6253

**Table 4 sensors-21-05666-t004:** RMSE values without and with L-CNN sentiment analysis model.

#	Amazon Fine Foods Reviews			Amazon Movie Reviews		
	SVD	NMF	SVD++	SVD	NMF	SVD++
Without sentiment	1.2076	1.2312	1.1831	0.9960	0.9464	0.9376
With sentiment (β = 0.3)	1.1338	1.2103	1.1292	0.8732	0.9112	0.8577
With sentiment (β = 0.5)	1.1442	1.2102	1.1356	0.8851	0.9041	0.8598
With sentiment (β = 0.7)	1.1633	1.2150	1.1493	0.9166	0.9110	0.8791

**Table 5 sensors-21-05666-t005:** NMAE values without and with L-CNN sentiment analysis model.

#	Amazon Fine Foods Reviews			Amazon Movie Reviews		
	SVD	NMF	SVD++	SVD	NMF	SVD++
Without sentiment	0.2427	0.2402	0.2385	0.2161	0.2022	0.2066
With sentiment (β = 0.3)	0.2091	0.2191	0.2066	0.1486	0.1484	0.1443
With sentiment (β = 0.5)	0.2158	0.2211	0.2117	0.1567	0.1494	0.1490
With sentiment (β = 0.7)	0.2233	0.2241	0.2177	0.1675	0.1531	0.1563

**Table 6 sensors-21-05666-t006:** MRR, MAP and NDCG values without and with L-CNN sentiment model on Amazon Fine Foods Reviews with β = 0.7.

#	SVD		NMF		SVD++	
	Without Sentiment	With Sentiment (*β* = 0.7)	Without Sentiment	With Sentiment (*β* = 0.7)	Without Sentiment	With Sentiment (*β* = 0.7)
MRR	83.92%	84.09%	82.98%	83.19%	84.24%	84.24%
MAP	73.06%	73.67%	72.97%	73.26%	73.48%	73.82%
NDCG	86.53%	86.78%	86.67%	86.84%	86.84%	86.89%

**Table 7 sensors-21-05666-t007:** MRR, MAP, and NDCG values without and with C-LSTM sentiment model on Amazon Fine Foods Reviews with β = 0.7.

#	SVD		NMF		SVD++	
	Without Sentiment	With Sentiment (*β* = 0.7)	Without Sentiment	With Sentiment (*β* = 0.7)	Without Sentiment	With Sentiment (*β* = 0.7)
MRR	83.92%	84.16%	82.98%	83.23%	84.24%	84.33%
MAP	73.06%	73.64%	72.97%	73.23%	73.48%	73.84%
NDCG	86.53%	86.78%	86.67%	86.82%	86.84%	86.89%

## Data Availability

Not applicable.

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
