# Peer review of "An Approach to Integrating Sentiment Analysis into Recommender Systems"

_sensors, 2021, doi:10.3390/s21165666_

Round 1
Reviewer 1 Report
The paper proposes a recommendation approach combining sentiment analysis with collaborative filtering. It is based on techniques for feature extraction and deep learning for the sentiment analysis, and incorporating sentiments into recommendations as additional feedback. The evaluation uses two popular datasets and show the proposed approach significantly improves the recommender system’s performance.
The considered problem is interesting and has theoretical and practical merit. The original contribution of the paper is limited, since for most of its part the paper is derivative, capitalizing on already established approaches, and already known approaches. There is no novelty infused in the employed methodologies. However, the provided results could be of interest for the corresponding community.
The presentation quality of the paper is overall fine and the text easy to follow. There are a few spots in the text with grammatical or syntactical mistakes that can be easily fixed with a careful proofreading. Some indicative examples: 1st and 2nd line of second paragraph in section 1, line 77, lines 150-151, line 281
The notation employed in eq. (1) regarding user a, and weight factor a is a bit confusing. It would be better to employed distinctive symbols for different parameters.
In line 283, a neighboring user of active user a is mentioned, but such neighborhood relationship has not been explained before. Does it regard distance with respect to similarity? But then how is the neighborhood of a user determined? This needs to be explicitly defined before used.
The problem studied and the proposed approach are well motivated, but there is an issue with content overlap in the first three sections. Literature review is split over sections 1 and 2. Then section 3 also presents background content on the basic two topics, namely sentiment analysis and recommender systems. It would be better to avoid this disruption in the flow of the content and present the whole of background and related work material as one, thus avoiding overlap and improving the structure of the paper.
In lines 49-50 there is a repetition that would be better to be removed.
Reviewer 2 Report
The manuscript aims at sentiment analysis into collaborative filtering methods by means of a recommender approach that is based on an adaptive architecture, which includes improved techniques for feature extraction and deep learning models based on sentiment analysis.
Limitations of the manuscript :
There are several typos and grammatical errors (punctuation, articles, etc.) and sentences at some instances in the manuscript with no clear meaning. Please revise the paper to fix them.
The manuscript needs a high-level and intuitive description of the concepts, methodologies in order to be able to follow them properly. In the current form, it’s too basic and unnecessary details are provided vaguely.
The baseline approach and comments regarding it are not mentioned in the manuscript.
It is not clear that what authors want to achieve by variation of CNN and LSTM (deep learning) networks and how actually they integrate BERT and the aforementioned.
The F-Score metrics used is micro or macro or else?
For holistically addressing the pre-processing strategy, word embedding andTF-IDF approaches please cite the recent manuscripts:
V. Kumar, D. R. Recupero, D. Riboni and R. Helaoui, "Ensembling Classical Machine Learning and Deep Learning Approaches for Morbidity Identification From Clinical Notes," in IEEE Access, vol. 9, pp. 7107-7126, 2021, DOI: 10.1109/ACCESS.2020.304322.
Dessi, D., Helaoui, R., Kumar, V., Reforgiato Recupero, D., & Riboni, D. (2020). Tf-IDF vs word embeddings for morbidity identification in clinical notes: An initial study. ACM SmartPhil 2020 (Vol. 2596, pp. 1-12). CEUR-WS. http://ceur-ws.org/Vol-2596/paper1.pdf
